# Associations of proton pump inhibitors with susceptibility to influenza, pneumonia, and COVID-19: Evidence from a large population-based cohort study

Ruijie Zeng[1,2,3†], Yuying Ma[1,2†], Lijun Zhang[1,4†], Dongling Luo[5†], Rui Jiang[1,4], Huihuan Wu[1,4], Zewei Zhuo[1], Qi Yang[1], Jingwei Li[1,4], Felix W Leung[6,7], Chongyang Duan[8*], Weihong Sha[1,2,3,4*], Hao Chen[1,2,3,4*]

[1]Department of Gastroenterology, Guangdong Provincial People's Hospital (Guangdong Academy of Medical Sciences), Southern Medical University, Guangzhou, China; [2]The Second School of Clinical Medicine, Southern Medical University, Guangzhou, China; [3]Shantou University Medical College, Guangdong, China; [4]School of Medicine, South China University of Technology, Guangzhou, China; [5]Guangdong Cardiovascular Institute, Guangdong Provincial People's Hospital, Guangdong Academy of Medical Sciences, Guangzhou, China; [6]David Geffen School of Medicine, University of California Los Angeles, Los Angeles, United States; [7]Sepulveda Ambulatory Care Center, Veterans Affairs Greater Los Angeles Healthcare System, North Hills, United States; [8]Department of Biostatistics, School of Public Health, Southern Medical University, Guangzhou, China

*For correspondence: donyduang@126.com (CD); shaweihong@gdph.org.cn (WS); chenhao@gdph.org.cn (HC)

†These authors contributed equally to this work

## Abstract

**Background:** Adverse effects of proton pump inhibitors (PPIs) have raised wide concerns. The association of PPIs with influenza is unexplored, while that with pneumonia or COVID-19 remains controversial. Our study aims to evaluate whether PPI use increases the risks of these respiratory infections.

**Methods:** The current study included 160,923 eligible participants at baseline who completed questionnaires on medication use, which included PPI or histamine-2 receptor antagonist (H2RA), from the UK Biobank. Cox proportional hazards regression and propensity score-matching analyses were used to estimate the hazard ratios (HRs) and 95% confidence intervals (CIs).

**Results:** Comparisons with H2RA users were tested. PPI use was associated with increased risks of developing influenza (HR 1.32, 95% CI 1.12–1.56) and pneumonia (hazard ratio [HR] 1.42, 95% confidence interval [CI] 1.26–1.59). In contrast, the risk of COVID-19 infection was not significant with regular PPI use (HR 1.08, 95% CI 0.99–1.17), while the risks of severe COVID-19 (HR 1.19. 95% CI 1.11–1.27) and mortality (HR 1.37. 95% CI 1.29–1.46) were increased. However, when compared with H2RA users, PPI users were associated with a higher risk of influenza (HR 1.74, 95% CI 1.19–2.54), but the risks with pneumonia or COVID-19-related outcomes were not evident.

**Conclusions:** PPI users are associated with increased risks of influenza, pneumonia, as well as COVID-19 severity and mortality compared to non-users, while the effects on pneumonia or COVID-19-related outcomes under PPI use were attenuated when compared to the use of H2RAs. Appropriate use of PPIs based on comprehensive evaluation is required.

**Funding:** This work is supported by the National Natural Science Foundation of China (82171698, 82170561, 81300279, 81741067, 82100238), the Program for High-level Foreign Expert Introduction

of China (G2022030047L), the Natural Science Foundation for Distinguished Young Scholars of Guangdong Province (2021B1515020003), the Guangdong Basic and Applied Basic Research Foundation (2022A1515012081), the Foreign Distinguished Teacher Program of Guangdong Science and Technology Department (KD0120220129), the Climbing Program of Introduced Talents and High-level Hospital Construction Project of Guangdong Provincial People's Hospital (DFJH201923, DFJH201803, KJ012019099, KJ012021143, KY012021183), and in part by VA Clinical Merit and ASGE clinical research funds (FWL).

## eLife assessment

This **useful** study aimed to quantify associations between regular use of proton-pump inhibitors (PPI) with the occurrence of respiratory infections, such as influenza, pneumonia, COVID-19, and others over a period of several years. PPI use was associated with increased risks of influenza, pneumonia, but not of COVID-19, although severity and mortality of COVID-19 infections were higher in PPI users. There are inevitable weaknesses of the study design used, such as the fact that PPI use was only measured at one time-point whereas infections were assessed over a long time period, but these are appropriately highlighted in the discussion. Weaknesses are highlighted in the discussion and the study presents **convincing** evidence for the conclusions overall.

## Introduction

Proton pump inhibitors (PPIs), as one of the most commonly used medications worldwide, have been utilized for treating various conditions related to excessive gastric acid secretion (*Vaezi et al., 2017*). In the United States, the prescription of PPIs has doubled from 1999 to 2012, and the number of people taking PPIs is still increasing due to their availability over the counter (*Kantor et al., 2015*). However, concerns are being raised regarding the long-term and inappropriate use of PPIs, which have been linked to a wide range of adverse conditions, including osteoporotic fractures, renal failure, and vitamin deficiencies (*Targownik et al., 2022*; *Abtahi et al., 2021*; *Xie et al., 2019*).

PPI-induced hypochlorhydria and gastrointestinal residence of pathogens might increase the risk of respiratory infections (*Malfertheiner et al., 2017*). Cohort studies in the United Kingdom and the United States reveal the risks of developing community- and hospital-acquired pneumonia are increased by approximately 100% and 30%, respectively (*Herzig et al., 2009*; *van der Sande et al., 2021*). In contrast, a nested case-control study based on the UK General Practice Research Database indicates long-term PPI therapy is not associated with increased risk for community-acquired pneumonia (*Sarkar et al., 2008*), and a retrospective cohort study involving 593,265 patients in Canada demonstrates no increased risk in developing pneumonia among PPI recipients (*Redelmeier et al., 2010*). Recently, attention has also been paid to the susceptibility to Coronavirus Disease 2019 (COVID-19) caused by the severe acute respiratory syndrome coronavirus 2 (SARS-CoV-2). Based on 53,130 participants in the United States, a dose-dependent increased risk of COVID-19 positivity among PPI users is found (*Almario et al., 2020*). Another Danish study also indicate a marginally increased risk of COVID-19 positivity (*Israelsen et al., 2021*), whereas studies based on the UK Biobank and a Korean cohort indicate nonsignificant association (*Fan et al., 2021*; *Lee et al., 2021*). Meta-analyses on the associations between the use of PPI and SARS-CoV-2 infection also demonstrate conflicting results (*Israelsen et al., 2021*; *Li et al., 2021*).

To date, the association between PPI and influenza remains unknown. The current evidence referring to PPI, pneumonia and COVID-19 is controversial. Previous studies had several limitations. For instance, the study based on the UK General Practice Research Database did not adjust for potential confounding variables for PPI indications (*Othman et al., 2016*). Direct comparisons with histamine-2 receptor antagonist (H2RA) users were not conducted in previous studies to minimize confounding by indication. In addition, the findings by Almario et al. were based on patients with gastrointestinal symptoms, rather than the general population (*Almario et al., 2020*). The previous UK Biobank study merely included participants tested for COVID-19 from March to June 2020 (*Fan et al., 2021*).

By leveraging the large-scale cohort and updated data in the UK Biobank, we aim to evaluate the associations between the regular use of PPIs and the susceptibility to respiratory infections, including influenza, pneumonia, and COVID-19.

## Methods

### Study population

The detailed information on study design for the UK Biobank was described previously (*Sudlow et al., 2015*). Invitations were sent to about 9.2 million people who were aged 40–69, had capacity to consent, registered with the National Health Service (NHS), and lived within 25 miles of one of the assessment centers (*Conroy et al., 2023*). The participants were free to withdraw at any time (*Sudlow et al., 2015*). Over 0.5 million participants were recruited from 22 assessment centers in Scotland, England, and Wales (specific locations of assessment centers are available at: https://biobank.ndph. ox.ac.uk/ukb/field.cgi?id=54) from 2006 to 2010. Information such as touch screen questionnaire, interview, blood pressure, eye measurements, physical measurements and so on was collected in the assessment centers (detailed content of assessments is available at: https://biobank.ndph.ox.ac.uk/ ukb/refer.cgi?id=100241). Written informed consent was acquired from each participant, and ethical approval was obtained from the North West Multi-Center Research Ethics Committee (approval number: 11/NW/0382, 16/NW/0274, and 21/NW/0157). The current study has been approved under the UK Biobank project 83339. In this study, 11,171 participants with missing PPI medication data and 56,907 participants with missing covariate data were excluded, and we further restricted the cohort to

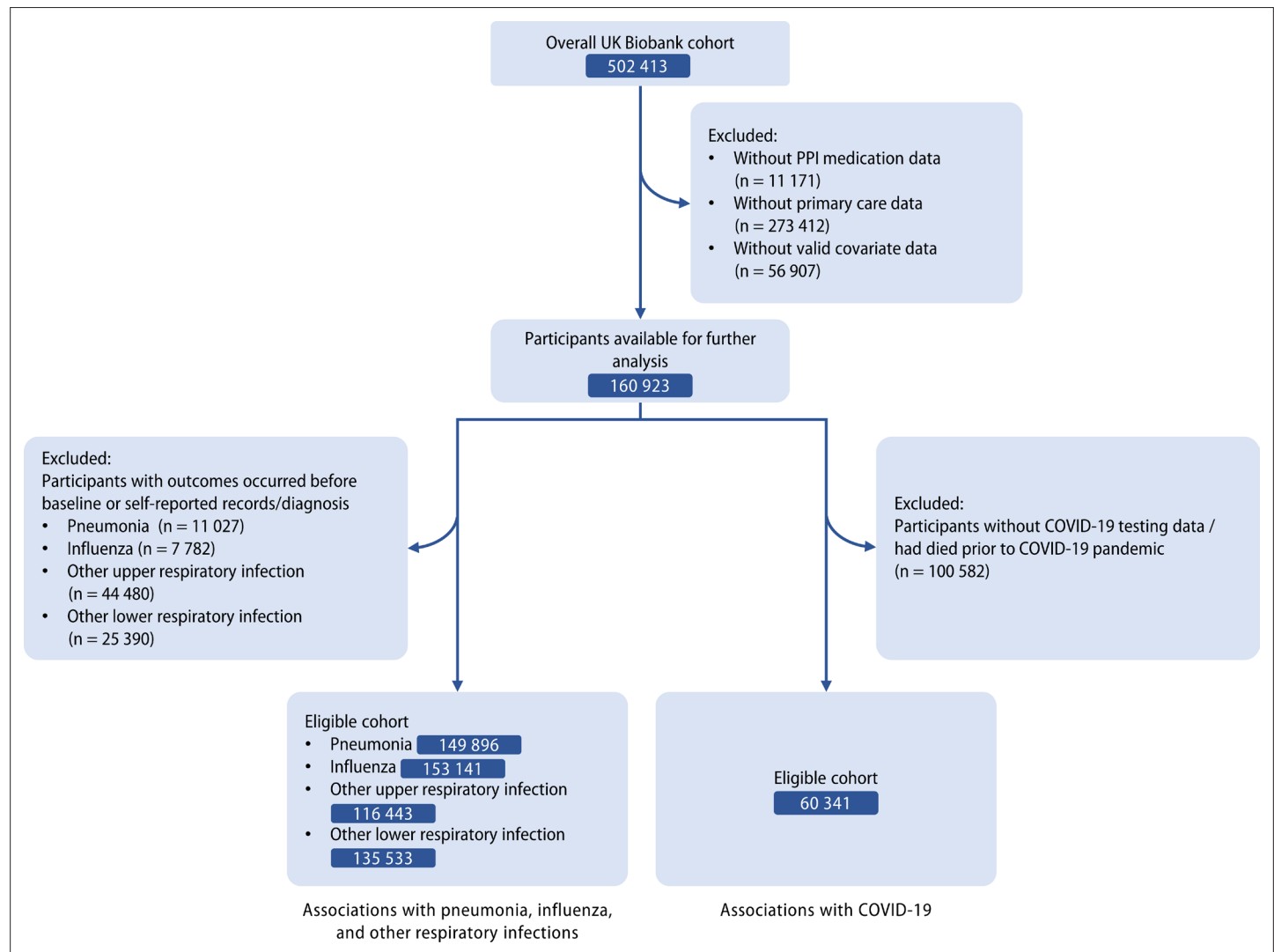

**Figure 1.** Flow diagram of eligible participants selection.

The online version of this article includes the following figure supplement(s) for figure 1:

**Figure supplement 1.** Directed acyclic graph (DAG) for evaluation of covariates in the logistic regression model.

the participants with available primary-care data. Among them, 1297 participants without follow-up, which were mainly determined by reported death, departure from the UK, or withdrawn consent, had been removed after initial exclusion. For the evaluation of associations with influenza, pneumonia, and other respiratory infections, those with outcomes occurred before the baseline, or only with self-reported records and diagnoses were further excluded. For the COVID-19 infection and COVID-19-related outcomes, we excluded participants whose COVID-19 testing data were unavailable or who had died before the COVID-19 pandemic (*Figure 1*).

## Definition of exposure

The exposure of interest was regular use of PPIs. The participants could enter the generic or trade name of the treatment on the touchscreen to match the medications they used (*Supplementary file 1a*). Verbal interviews were conducted by a trained nurse if participants answered that they were regularly taking prescribed or over-the-counter medication on the touchscreen, in which 'regular' was defined as most days of the week for the last 4 weeks, and information on specific types of medications was further recorded, while no response to the question on the interview was considered missing data for PPI use. Short-term medications, for example, a 1 week course of antibiotics, were not recorded in the interview. Types of PPIs available in the UK Biobank included omeprazole, lansoprazole, esomeprazole, rabeprazole, and pantoprazole. The regular use of H2RAs was also defied by the above process. When comparing PPI users with H2RA users, participants who took both medications were excluded. Information on dose or duration of acid suppressant use was not available in the UK Biobank.

## Definition of outcomes

The primary outcomes of interest were influenza, pneumonia, COVID-19 infections (*Supplementary file 1b*). Briefly, the first reported occurrences of respiratory system-related conditions within primary care data and hospital inpatient data defined by the International Classification of Diseases (ICD)- 10 codes were categorized by the UK Biobank (https://biobank.ndph.ox.ac.uk/showcase/label.cgi?id=2410). Influenza included those caused by identified influenza virus (J09-J10) and virus not identified (J11). Pneumonia was defined as that caused by viruses (J12), bacteria (J13-15), and other infectious organisms (J16-18).

COVID-19-related data in the UK Biobank (available from January 2020 to September 2021) based on follow-up of the participants was used (*Armstrong et al., 2020*). COVID-19 infection mainly included information on positive COVID-19 tests, and patients with inpatient diagnoses or mortality due to COVID-19 were also regarded as having COVID-19 infection.

The secondary outcomes included other upper or lower respiratory infections, COVID-19 mortality, and COVID-19 severity. The definition of other upper respiratory infections contained acute nasopharyngitis, sinusitis, pharyngitis, tonsillitis, laryngitis, tracheitis, obstructive laryngitis, epiglottitis, or upper respiratory infections of multiple and unspecified sites (J00-J06). Other lower respiratory infections included acute bronchitis, bronchiolitis, and other unspecified ones (J20-J22). Severe COVID-19 cases were defined as being hospitalized for COVID-19. COVID-19 mortality was defined as the underlying recorded cause of death due to COVID-19 (ICD-10 U07.1 and U07.2).

## Assessment of covariates

The covariates used for adjustments in our study were identified by a directed acyclic graph (DCA, *Figure 1—figure supplement 1*) based on existing literature and expert knowledge. Baseline data on sociodemographic information (age, sex, ethnicity), socioeconomic status (deprivation index, which was defined using national census information on car ownership, household overcrowding, owner occupation, and unemployment combined for postcode areas of residence), alcohol consumption, smoking status, fresh fruit intake, multivitamin use, and body mass index (BMI) were collected from the UK Biobank, while physical activity was assessed using the International Physical Activity Questionnaire-Short Form. Gastroesophageal reflux disease (GERD), peptic ulcers, and upper gastrointestinal bleeding, were included as they are main indications for the use of PPIs. The comorbidities (hypertension, type 2 diabetes, renal failure, myocardial infarction, stroke, chronic obstructive pulmonary disease [COPD], asthma) were examined using self-reported data and adjusted due to their impact on the risk of respiratory infections. Since PPI and H2RA have highly similar indications, the

use of H2RA was also adjusted. Data on medication use including aspirin, non-aspirin non-steroidal anti-inflammatory drugs (NSAIDS, including ibuprofen), and cholesterol-lowering medications were extracted and adjusted. For influenza and COVID-related outcomes, vaccinations were additionally adjusted. The linearity between continuous variables and outcomes was assessed by Martingale residuals plots, while the variables detected with non-linearity were regarded as categorical variables for further analyses.

### CYP2C19 genetic variants

PPIs are mainly cleared by *CYP2C19*, and therefore their metabolism and effects are affected by different variants of *CYP2C19*. Genotyped genetic variant data after quality control was available for UK Biobank participants based on the Affymetrix Axiom UKB array and the Affymetrix UKBiLEVE array (*Bycroft et al., 2018*). According to the Clinical Pharmacogenetics Implementation Consortium Guideline for *CYP2C19* and Proton Pump Inhibitor Dosing (*Lima et al., 2021*), genotypic data of four *CYP2C19* variants, including *rs12248560 (CYP2C19\*17)*, *rs17884712 (CYP2C19\*9)*, *rs4986893 (CYP2C19\*3)*, and *rs4244285 (CYP2C19\*2)*, were utilized to divide PPI users into three subgroups: (1) *CYP2C19* rapid and ultrarapid metabolizers (carried 1 functional allele and 1 increased-function allele [\*17]; or carried 2 increased-function alleles); (2) *CYP2C19* normal metabolizers (carried 2 functional alleles); (3) *CYP2C19* likely intermediate, intermediate and poor metabolizers (carried ≥1 alleles with no/decreased function [\*2, \*3, and \*9]).

### Statistical analysis

The baseline characteristics were demonstrated by percentages for categorical variables, and mean (standard deviation [SD]), or median (interquartile range [IQR]) for continuous variables according to the distribution of data after evaluating the data distribution.

Univariate and multivariable Cox proportional hazards regression models were utilized to assess the association between regular use of PPIs and the selected outcomes, and the results were presented as hazard ratios (HRs) with 95% confidence intervals (CIs). Multivariable model 1 included age and sex. Model 2 additionally contained other potential confounders selected a priori, including ethnicity, deprivation index, alcohol consumption, smoking, physical activity, BMI, fresh fruit intake, GERD, peptic ulcer, upper gastrointestinal bleeding, hypertension, type 2 diabetes, renal failure, myocardial infarction, stroke, COPD, asthma, aspirin, non-aspirin NSAIDS, H2RA, cholesterol-lowering medications, and multivitamin use. The reference group was participants without regular use of PPIs. Schoenfeld residuals tests were used to evaluate the proportional hazards assumptions, while no violation of the assumption was detected (*Supplementary file 1c*). Person-years were calculated from the number of participants and the date from January 2020 (for COVID-19-related outcomes) or recruitment (for other respiratory outcomes) to outcome diagnosis, last follow-up (September 2021 for COVID-19 infection and related outcomes; December 2021 for other outcomes), or death, whichever came first. Stratified analyses according to population characteristics, types of PPIs, and *CYP2C19* metabolizers were performed using multivariable-adjusted models across subgroups of each stratifying variable, and the multiplicative interactions were evaluated using likelihood ratio tests.

Quantitative bias analyses were performed to calculate e-values, which illustrates the strength of association between an unmeasured confounder and exposure or outcome, conditional on the measured covariates (*VanderWeele and Ding, 2017*). E-value is the smallest magnitude of risk estimates that an unmeasured confounder would need to have with the exposure and outcome to explain away an observed association (*VanderWeele and Ding, 2017*). The event-free probabilities were compared by Kaplan-Meier survival curves with inverse probability weights adjusting for the measured covariates. In addition, we conducted additional analyses using multiple imputation by chained equations to include participants initially excluded due to missing ethnicity data using the 'mice' package (*van Buuren and Groothuis-Oudshoorn, 2011*). Self-reported infections, except for COVID-19-related outcomes due to the lack of data, were also included for the outcomes as sensitivity analyses. The self-reported cases were reported at the baseline or subsequent UK Biobank assessment center visit. Moreover, propensity score-matching analysis was conducted. The same set of covariates was used to derive propensity scores, and the PPI users and non-users were matched with a ratio of 1:4 using the 'MatchIT' package (*Stuart et al., 2011*), which estimated the propensity scores in the background and matched observations based on the nearest neighbor method. The remaining imbalanced covariates

(standardized mean difference ≥0.1) after propensity score matching were further adjusted by multi-variable Cox regression models to calculate HRs and 95% CIs (*Normand et al., 2001*). Furthermore, because PPI and H2RA share highly similar indications, we performed head-to-head comparisons between PPI and H2RA users to further minimize the protopathic bias (*Abrahami et al., 2022a*; *Abrahami et al., 2022b*).

All statistical analyses were performed using R (*R Development Core Team, 2022*). The significance level at $\alpha$=0.05 with two tails was used. Risk estimates were reported with 95% CIs.

## Results

### Study population

A total of 160,923 individuals aged 38–71 years who passed the initial selection criteria in the UK Biobank were included in this study (*Table 1*). The median follow-up was 7.1 (interquartile range [IQR] 6.2–8.5) years. The mean age of the included participants was 56.5 years, and 53.0% of them were women. Evidently, regular PPI users were characterized by higher rates of GERD (32.4% vs 2.7%), peptic ulcer (5.6% vs 0.9%), and upper gastrointestinal bleeding (0.2% vs 0.03%) compared to non-regular PPI users. Higher burdens of comorbidities, as well as increased use of aspirin, H2RA, and cholesterol-lowering medications, were also observed in regular PPI users.

### Proton pump inhibitor use and influenza, pneumonia, and COVID-19 infection

Increased risks of developing influenza, pneumonia, and other respiratory infections were identified in regular users of PPIs compared with non-regular users, and the risk remained raised after adjustments (*Table 2*). Inclusion of the self-reported cases did not significantly alter the results (*Supplementary file 1d*). A 32% increased risk of developing influenza (aHR 1.32, 95% CI 1.12–1.56, p=0.001; e-value 1.97) was observed among regular PPI users. In addition, regular use of PPIs was associated with a 42% increased risk of developing pneumonia (fully adjusted HR [aHR] 1.42, 95% CI 1.26–1.59, p<0.001; e-value 2.19). Regular PPI users had lower event-free probabilities for influenza and pneumonia compared to those of non-users (*Figure 2—figure supplement 1A-B*). The association of PPI use with COVID-19 positivity was further evaluated in our study. Initially, in the non-adjusted model, the susceptibility to COVID-19 positivity was observed with a 18% increase (HR 1.18, 95% CI 1.09–1.26, p<0.001 for non-adjusted model; *Table 2*) in participants with regular use of PPIs. However, full adjustments for covariates rendered the association nonsignificant (aHR 1.08, 95% CI 0.99–1.17, p=0.101; *Table 2*).

### Proton pump inhibitor use and other respiratory infections, COVID-19 severity, and COVID-19 mortality

For other upper and lower respiratory infections, the risks among regular PPI users were increased by 19% (aHR 1.19, 95% CI 1.11–1.27, p<0.001; e-value 1.67) and 37% (aHR 1.37, 95% CI 1.29–1.46, p<0.001; e-value 2.08), respectively. In contrast, the risks of developing severe COVID-19 (aHR 1.33, 95% CI 1.09–1.61, p=0.004; e value 1.99) and mortality due to COVID-19 (aHR 1.46, 95% CI 1.05–2.03, p=0.024; e value 2.03) were significantly increased among PPI users compared to those among PPI non-users (*Supplementary file 1e*). PPI users had lower event-free probabilities for COVID-19 severity and mortality, but not COVID-19 positivity compared to those of non-users (*Figure 2—figure supplement 1C—E*).

### Subgroup analysis

Stratified analyses were performed in the fully adjusted models for the main outcomes. Overall, no significant evidence of interactions was observed in the subgroup analyses referring to influenza (all p for interaction >0.05, *Figure 2*). The subgroup analyses for other main outcomes were illustrated in *Figure 2* and *Figure 2—figure supplement 2*.

Among different types of PPIs, regular omeprazole or lansoprazole users were correlated with greater risks of respiratory infections (*Supplementary file 1f*). The risks of influenza were significant among *CYP2C19* normal metabolizers, and the risk estimate increased among *CYP2C19* likely intermediate, intermediate and poor metabolizers, while more information and larger sample sizes on

**Table 1.** Baseline characteristics of the included participants.

| Characteristics | Regular PPI use | | |
| --- | --- | --- | --- |
| | Yes | No | Overall |
| Number of participants, n (%) | 9 997 (6.2) | 150 926 (93.8) | 160 923 (100.0) |
| Age, years, mean (SD) | 59.4 (7.4) | 56.3 (8.2) | 56.5 (8.1) |
| Sex, female, n (%) | 5 533 (55.4) | 79 709 (52.8) | 85 242 (53.0) |
| Ethnicity, white, n (%) | 9 571 (95.7) | 144 295 (95.6) | 153 866 (95.6) |
| Deprivation index, mean (SD) | –0.9 (3.3) | –1.4 (3.0) | –1.4 (3.0) |
| Alcohol consumption, n (%) | | | |
| Daily or almost daily | 1 805 (18.1) | 28 846 (20.5) | 32 874 (20.4) |
| 3 or 4 times a week | 1 923 (19.2) | 33 533 (23.8) | 37 570 (23.4) |
| 1 or 2 times a week | 2 396 (24.0) | 37 443 (26.6) | 42 228 (26.2) |
| 1–3 times a month | 1 179 (11.8) | 15 669 (11.1) | 17 988 (11.2) |
| Special occasions only | 1 522 (15.2) | 14 965 (10.6) | 17 616 (11.0) |
| Never | 1 161 (11.6) | 10 414 (7.4) | 12 570 (7.8) |
| Smoking, n (%) | | | |
| Never smoker | 4 572 (45.7) | 82 998 (55.0) | 87 570 (54.4) |
| Previous smoker | 4 289 (42.9) | 52 013 (34.5) | 56 302 (35.0) |
| Current smoker | 1 136 (11.4) | 15 915 (10.6) | 17 051 (10.6) |
| Physical activity, MET minutes/week, median (IQR) | 1 525.5 (2 722.0) | 1 815.0 (2 848.5) | 1 794.0 (2 838.5) |
| Fresh fruit intake, pieces, mean (SD) | 2.0 (2.6) | 1.9 (2.6) | 1.9 (2.6) |
| BMI, kg/m2, mean (SD) | 29.2 (5.1) | 27.4 (4.7) | 27.5 (4.8) |
| Indication of PPIs, n (%) | | | |
| GERD | 3 235 (32.4) | 4 015 (2.7) | 7 250 (4.5) |
| Peptic ulcer | 561 (5.6) | 1 303 (0.9) | 1 864 (1.2) |
| Upper gastrointestinal bleeding | 18 (0.2) | 38 (0.03) | 56 (0.03) |
| Comorbidities, n (%) | | | |
| Hypertension | 4 116 (41.2) | 38 162 (25.3) | 42 278 (26.3) |
| Type 2 diabetes | 124 (1.2) | 890 (0.6) | 1 014 (0.6) |
| Renal failure | 60 (0.6) | 243 (0.2) | 303 (0.2) |
| Myocardial infarction | 331 (3.3) | 1 632 (1.1) | 1 963 (1.2) |
| Stoke | 140 (1.4) | 943 (0.6) | 1 083 (0.7) |
| COPD | 46 (0.5) | 200 (0.1) | 246 (0.2) |
| Asthma | 841 (8.4) | 8 471 (5.6) | 9 312 (5.8) |
| Medication use, n (%) | | | |
| Aspirin | 2 457 (24.6) | 21 108 (14.0) | 23 565 (14.6) |
| Non-aspirin NSAIDS | 1 224 (12.2) | 22 568 (15.0) | 23 792 (14.8) |
| H2RA | 297 (3.0) | 2 956 (2.0) | 3 253 (2.02) |
| Cholesterol lowering medications | 1 537 (15.4) | 9 241 (6.1) | 10 778 (6.70) |
| Multivitamin use, n (%) | 2 227 (22.3) | 33 201 (22.0) | 35 428 (22.0) |

BMI: body mass index; COPD: chronic obstructive pulmonary disease; GERD: gastroesophageal reflux disease; H2RA: histamine 2 receptor antagonist; IQR: interquartile range; MET: metabolic equivalent of task; PPI: proton pump inhibitor; NSAIDS: non-steroidal anti-inflammatory drugs; SD: standard deviation.

**Table 2.** Associations of PPI use with the susceptibility to pneumonia, influenza, COVID-19 positivity, and other respiratory infections.

| | Case/person-years | Non-adjusted model | | Age/sex-adjusted model | | Fully adjusted model* | |
|---|---|---|---|---|---|---|---|
| | | HR (95% CI) | p | HR (95% CI) | p | HR (95% CI) | p |
| Influenza | | | | | | | |
| Non-regular PPI use | 2 009/6 011 | 1.00 (reference) | | 1.00 (reference) | | 1.00 (reference) | |
| Regular PPI use | 183/539 | 1.38 (1.19–1.62) | <0.001 | 1.49 (1.28–1.74) | <0.001 | 1.32 (1,12–1.56) | 0.001 |
| Pneumonia | | | | | | | |
| Non-regular PPI use | 2 904/12 867 | 1.00 (reference) | | 1.00 (reference) | | 1.00 (reference) | |
| Regular PPI use | 378/1 702 | 2.04 (1.83–2.27) | <0.001 | 1.74 (1.56–1.94) | <0.001 | 1.42 (1.26–1.59) | <0.001 |
| COVID-19 positivity | | | | | | | |
| Non-regular PPI use | 23 989/29 080 | 1.00 (reference) | | 1.00 (reference) | | 1.00 (reference) | |
| Regular PPI use | 1 440/1 702 | 1.18 (1.09–1.26) | <0.001 | 1.07 (0.99–1.15) | 0.058 | 1.08 (0.99–1.17) | 0.101 |
| Other upper respiratory infections | | | | | | | |
| Non-regular PPI use | 14 449/52 499 | 1.00 (reference) | | 1.00 (reference) | | 1.00 (reference) | |
| Regular PPI use | 1 118/3 988 | 1.30 (1.22–1.38) | <0.001 | 1.31 (1.23–1.39) | <0.001 | 1.19 (1.11–1.27) | <0.001 |
| Other lower respiratory infections | | | | | | | |
| Non-regular PPI use | 14 494/55 384 | 1.00 (reference) | | 1.00 (reference) | | 1.00 (reference) | |
| Regular PPI use | 1 486/5 598 | 1.78 (1.67–1.88) | <0.001 | 1.65 (1.56–1.74) | <0.001 | 1.37 (1.29–1.46) | <0.001 |

CI: confidence interval; COVID-19: coronavirus disease 2019; HR: hazard ratio; PPI: proton pump inhibitor.

*Adjusted for age, sex, ethnicity, deprivation index, smoking, alcohol consumption, physical activity, fresh fruit intake, body mass index, any indication of PPIs (gastroesophageal reflux disease [GERD], peptic ulcer, upper gastrointestinal bleeding), comorbidities (hypertension, type 2 diabetes, renal failure, myocardial infarction, stroke, chronic obstructive pulmonary disease [COPD], asthma), medications (aspirin, non-aspirin non-steroidal anti-inflammatory drugs [NSAIDs, ibuprofen], histamine 2 receptor antagonists (H2RAs), cholesterol lowering medications), multivitamin use, and influenza vaccination (for influenza) or COVID-19 vaccination (for COVID-19-related outcomes).

PPI subtypes are still needed to increase the statistical power (*Supplementary file 1g*). The risks of COVID-19 severity and COVID-19 mortality were higher among *CYP2C19* likely intermediate, intermediate and poor metabolizers (*Supplementary file 1h*). The risks of pneumonia were higher among *CYP2C19* rapid and ultrarapid metabolizers (*Supplementary file 1g*).

## Analysis by multiple imputation and propensity score-matching

After imputation of missing data, we found that individuals with regular use of PPIs were associated with similarly increased trends in the risks of influenza, pneumonia, other upper respiratory infections, and other lower respiratory infections (all p<0.05; *Supplementary file 1i*). The associations with COVID-19 severity and mortality were also significant (all p<0.05; *Supplementary file 1j*).

Matching of 9910 regular PPI users and 39,760 non-regular users (1:4 by propensity scores) was also conducted, and the baseline characteristics were much more similar (*Supplementary file 1k*). The participants regularly exposed to PPIs were observed with increased risks for influenza, pneumonia, other upper respiratory infections, and other lower respiratory infections (all p<0.05; *Supplementary file 1l*), which were consistent with the results from Cox hazard proportional regression models. The associations with COVID-19 severity and mortality were also significant (all p<0.05; *Supplementary file 1m*).

## Comparisons with H2RA users

To further confirm the results and reduce the effect of confounding by indications, we evaluated the risk of respiratory infections compared to the use of H2RAs, which is a less potent acid-suppressant and contains indications similar to PPI. When compared to regular H2RA users, participants with regular use of PPIs were also associated with an increased risk of influenza (HR 1.74, 95% CI 1.19–2.54, p=0.004; e-value 2.87), other upper respiratory infection (HR 1.28, 95% CI 1.07–1.54, p=0.008;

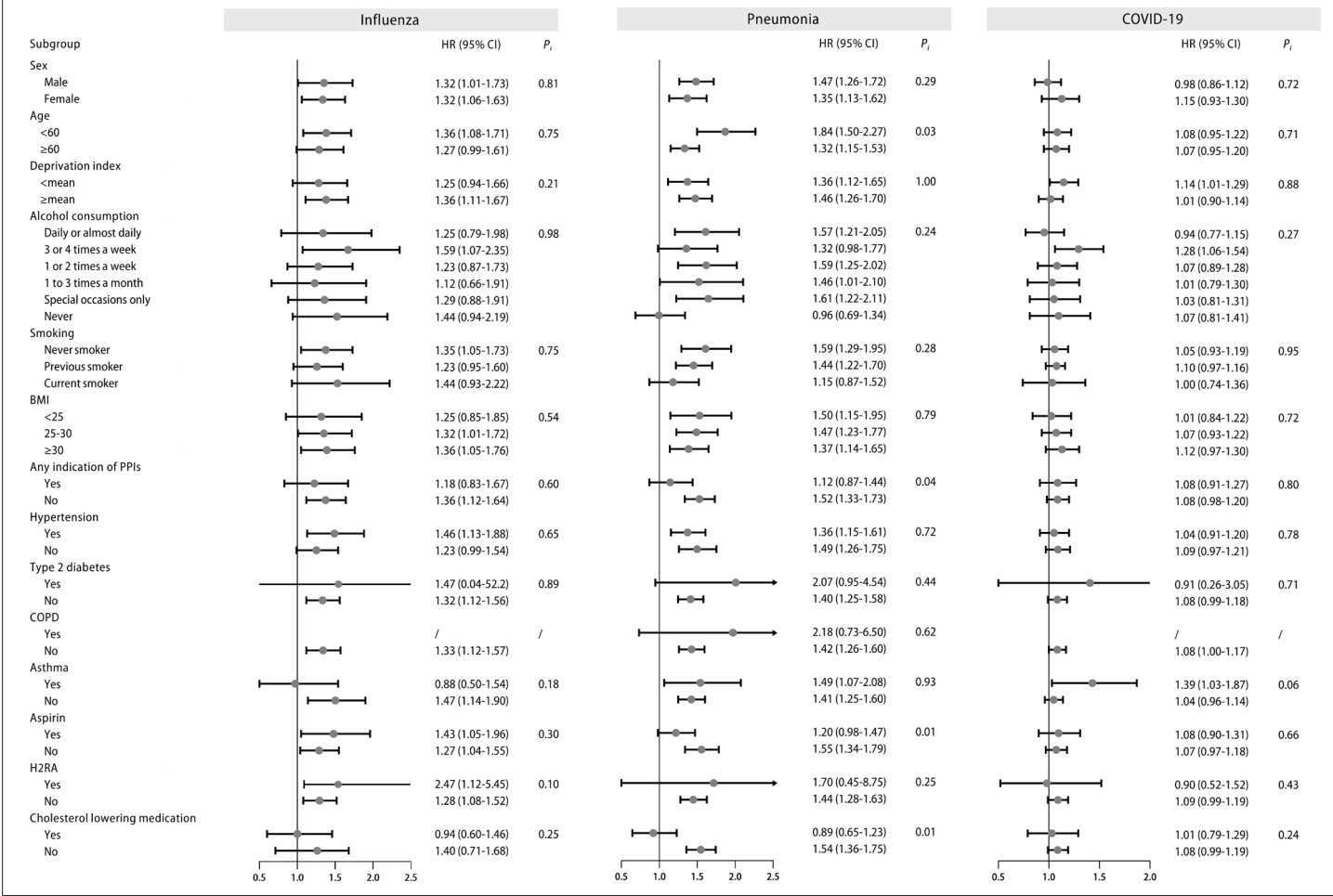

**Figure 2.** Stratified analysis of regular proton pump inhibitor (PPI) users and the risk of influenza, pneumonia, and COVID-19 infection. Effect estimates were based on age, sex, deprivation index, alcohol consumption, smoking, body mass index (BMI), indications of PPIs, hypertension, type 2 diabetes, chronic obstructive pulmonary disease (COPD), asthma, aspirin, histamine 2 receptor antagonist (H2RA), and cholesterol-lowering medication, using the fully adjusted model. CI: confidence interval; HR: hazard ratio; $P_i$: *P* value for interaction.

The online version of this article includes the following figure supplement(s) for figure 2:

**Figure supplement 1.** Kaplan-Meier curves illustrating the event-free probability for the outcomes among users and non-users of proton pump inhibitors (PPIs).

**Figure supplement 2.** Stratified analysis of proton pump inhibitor (PPI) users and the risk of COVID-19 severity and mortality.

**Figure supplement 3.** Directed acyclic graph (DAG) for potential unmeasured confounders.

e-value 1.88), and other lower respiratory infection (HR 1.33, 95% CI 1.18–1.50, p<0.001; e-value 1.99; *Table 3*). However, the associations with pneumonia (HR 1.22. 95% CI 0.96–1.54, p=0.104),COVID-19 infection (HR 1.04. 95% CI 0.87–1.26, p=0.629), COVID-19 severity (HR 0.91. 95% CI 0.64–1.30, p=0.608), or COVID-19 mortality (HR 0.83. 95% CI 0.45–1.56, p=0.745) were not significant (*Supplementary file 1n*).

## Discussion

In this large-scale, population-based cohort with updated information, we identify that the use of PPIs is associated with incident influenza. In contrast, analyses of pneumonia, COVID-19 infection and related outcomes, reveal attenuated effects after being compared with H2RA users. The association with influenza remains robust across different subgroups stratified by population characteristics and *CYP2C19* phenotypes.

**Table 3.** Comparisons of the risks of influenza, pneumonia, and COVID-19 between proton pump inhibitor (PPI) and histamine-2 receptor antagonist (H2RA) users.

| | Cases / Person-years | HR (95% CI)* | p |
|---|---|---|---|
| Influenza | | | |
| Regular H2RA use | 32/102 | 1.00 (Reference) | |
| Regular PPI use | 175/524 | 1.74 (1.19–2.54) | 0.004 |
| Pneumonia | | | |
| Regular H2RA use | 86/385 | 1.00 (Reference) | |
| Regular PPI use | 368/1653 | 1.22 (0.96–1.54) | 0.104 |
| COVID-19 positivity | | | |
| Regular H2RA use | 425/506 | 1.00 (Reference) | |
| Regular PPI use | 1 409/1 665 | 1.06 (0.90–1.24) | 0.509 |
| Other upper respiratory infection | | | |
| Regular H2RA use | 146/522 | 1.00 (Reference) | |
| Regular PPI use | 602/2099 | 1.28 (1.07–1.54) | 0.008 |
| Other lower respiratory infection | | | |
| Regular H2RA use | 339/1350 | 1.00 (Reference) | |
| Regular PPI use | 1438/5398 | 1.33 (1.18–1.50) | <0.001 |

CI: confidence interval; COVID-19: coronavirus disease 2019; H2RA: histamine-2 receptor antagonist; HR: hazard ratio; PPI:585 proton pump inhibitor.

*Adjusted for age, sex, ethnicity, deprivation index, smoking, alcohol consumption, physical activity, fresh fruit intake, body mass index, any indication of PPIs (gastroesophageal reflux disease [GERD], peptic ulcer, upper gastrointestinal bleeding), comorbidities (hypertension, type 2 diabetes, renal failure, myocardial infarction, stroke, chronic obstructive pulmonary disease [COPD], asthma), medications (aspirin, non-aspirin non-steroidal anti-inflammatory drugs [NSAIDs, ibuprofen], cholesterol lowering medications), multivitamin use, and influenza vaccination (for influenza) or COVID-19 vaccination (for COVID-19-related outcomes).

The correlation between PPI use and the risk of influenza remains unexplored. For the past two decades, accumulating evidence indicates increased risks of pneumonia under the use of PPIs (*Herzig et al., 2009*; *van der Sande et al., 2021*; *Meijvis et al., 2011*; *Jeon and Kim, 2022*), whereas others failed to show such associations (*Sarkar et al., 2008*; *Redelmeier et al., 2010*). Conflicting findings also exist for studies referring to the risk of COVID-19 infection and related outcomes among PPI users, including several meta-analyses (*Almario et al., 2020*; *Israelsen et al., 2021*; *Fan et al., 2021*; *Lee et al., 2021*; *Li et al., 2021*; *Shah et al., 2022*; *Wu et al., 2022*; *Shupp et al., 2022*; *Shafrir et al., 2022*). Compared with existing studies, our study more comprehensively adjusts for a variety of critical covariates by utilizing the latest data from the UK Biobank. In addition, distinct from previous population-based studies, we compared the risks with H2RA users to further reduce protopathic and other unmeasured bias, since the users of acid suppressants, including PPIs and H2RAs, can have matched information on different characteristics, including indications. Although the risks of pneumonia were initially increased in Cox and propensity-score-matched analyses, direct comparison with H2RA users showed negative results, which indicates that previously observed associations could be due to unmeasured confounders.

Several proposed mechanisms can account for the association between the use of PPIs and respiratory tract infections. Since a low pH of gastric acid rapidly inactivates microorganisms, one critical issue is that reduced acidity induced by PPIs leads to the overgrowth of microorganisms, which can contribute to the development of infections in the respiratory tract through microaspiration (*Scholtissek, 1985*). Colonization and growth of pathogens under hypochlorhydria could increase the risk of respiratory infections. Although initial assessments indicated the use of PPIs might increase the risk of pneumonia, the head-to-head comparison with H2RAs yielded impacted effects. It could be due

to the similar acid-suppressive effects of H2RAs and reduced sample size, which therefore warrants further investigations.

Concerns over protopathic bias due to non-specific and overlapping symptoms between influenza/pneumonia and acid-related diseases were raised (*Horwitz and Feinstein, 1980*). Nevertheless, pneumonia and influenza often present with acute cough, and other concomitant symptoms distinct from acid-related diseases (*Irwin et al., 2018*). In contrast, patients with chronic cough are more commonly GERD-related (*Irwin et al., 2018*). The American College of Chest Physicians Clinical Practice Guidelines for Management of Reflux-Cough Syndrome also recommend against using PPI therapy alone for patients with chronic cough but without heartburn or regurgitation (*Kahrilas et al., 2016*). In our study, the use of PPIs is defined as taking the medication for most days of the week in the last 4 weeks, which is uncommon for acute cough. Although we cannot completely rule out protopathic bias, we have attempted to minimize it by adjusting for covariates including PPI indications, matching with propensity scores, and comparing with H2RA users.

For the risk of developing influenza, we analyzed the risks among different *CYP2C19* metabolizers for the first time, and further observed a significant increase among *CYP2C19* normal metabolizers compared to rapid and ultrarapid metabolizers. Although the risks of several outcomes, for example, influenza and pneumonia, for *CYP2C19* likely intermediate, intermediate and poor metabolizers are not statistically significant, they could be due to the limited sample size, and the risk estimates are higher compared to those among other types of metabolizers. Intriguingly, the risks of developing influenza and pneumonia are higher among *CYP2C19* rapid and ultrarapid metabolizers regularly taking PPIs compared to other types of metabolizers. Since our study exclusively involves participants with valid primary care data, such an increased risk might be to some extent contributed by the over-prescription or self-taking of PPIs under the undesired effects without following the proper strategy. Our findings are generally consistent with the assumption that slower metabolizers are associated with higher risks of adverse effects, while larger samples are needed to increase statistical power. Prescription of PPIs based on different *CYP2C19* metabolism subtypes is therefore important to reduce the adverse effects.

Our study has several strengths. First, our study utilizes the updated large-sample data from the UK Biobank and exclusively includes participants with valid records from primary care, which reduces the information bias. Second, a variety of covariates, especially for the indications of PPIs and the use of aspirin, which might contribute to indication or protopathic bias, have been adjusted to enhance the robustness of our results. Third, genotypic data of metabolic enzymes has been integrated into our study. Fourth, propensity score-matching analysis reduces a greater portion of bias, and analyses by propensity-score matching or multiple imputation derive consistent results. Fifth, adjustments for vaccination for COVID-19 and influenza has been performed in our study to reduce the confounding effects by vaccination. Furthermore, the comparison with participants using H2RA, a less potent acid suppressant with similar indications, further reduces the confounding by indication. The findings on the risk of influenza remain highly consistent across different strata and sensitivity analyses.

Limitations exist in our study. Information on dose and duration of PPI use, discrimination between prescription and over-the-counter use of PPIs, health-seeking behavior, different types of pneumonia, and pneumococcus vaccination is currently not available from the UK Biobank. Given that the PPI exposure was mainly assessed at the baseline recruitment, it was possible that a small proportion of PPI users was misclassified during the follow-up due to the medication discontinuation, which may result in an underestimation of potential risk. However, the prevalent user design could underestimate the actual risks of PPI use for respiratory infections, which indicates the real effect might be stronger (*Fu et al., 2021*). In addition, no effect moderation was observed in subgroup analyses for the main outcome among PPI users with indications (more likely to regularly use PPIs for a long period) compared to those without indications, indicating the risks remained increased among long-term PPI users. Since the follow-up prescription data was lacking in our study to precisely identifying the long-term users, further evaluation using cohorts with close follow-up is needed. PPIs are indicated for *Helicobacter pylori* eradication, whereas the UK Biobank does not contain adequate data. Thus, the indication for eradicating *H. pylori* is not adjusted in this study. The data on different PPI subtypes and COVID-19 infection and related outcomes are relatively small, which limits their power and still needs further investigation. Moreover, patients with exacerbations of comorbid disorders (e.g. diabetes, asthma, COPD) might suffer from a wide range of gastrointestinal symptoms that lead to the use of

PPIs (*Etminan et al., 2021*; *Figure 2—figure supplement 3*). Due to the lack of data for respiratory severity and close follow-up for medication use, residual confounding might still exist due to the observational nature. Residual genotyping impacts of other enzymes, although affecting the metabolism to a lesser extent, might also exist. Although no significant differences were found between PPIs and H2RAs regarding the association with pneumonia and COVID-19-related outcomes, this could be due to the reduced sample size and power, which require larger cohorts to validate the effects. Furthermore, the highly selective nature of the UK Biobank might create collider stratification bias for the evaluation of COVID-19-related outcomes, and thus the conclusions should be interpreted with cautions (*Griffith et al., 2020*). The current study is principally based on white British ancestry in the United Kingdom, and future exploration of other ancestries with comparisons is warranted.

Our findings could have essential implications for the prevention of respiratory infections and the de-prescribing of PPIs in clinical practice. Administration of PPIs can rapidly increase intragastric pH to higher than 6 after 2–4 hours (*Laine et al., 2008*). Emerging evidence has revealed the inappropriate prescription of PPIs in both the primary and secondary care settings, and 33–67% of the patients did not take the drug according to their countries' criteria (*Forgacs and Loganayagam, 2008*). Similarly, the baseline characteristics of the included participants in our study demonstrate that approximately 60% of the regular PPI users do not have main indications. In addition, although influenza is usually self-limiting in healthy individuals, its risk of complications is significantly increased among pregnant women and people with immunosuppression or chronic diseases (*Ghebrehewet et al., 2016*). Therefore, comprehensive evaluation of PPI use is needed in clinical practice.

## Conclusion

In conclusion, compared to non-users, people regularly taking PPIs are associated with increased susceptibility to influenza, pneumonia, as well as COVID-19 severity and mortality, while their association with pneumonia and COVID-19-related outcomes is diminished after comparison with H2RA use and remains to be further explored.

## Acknowledgements

The authors are grateful to the UK Biobank for approval and access to data of the project, and this research has been conducted under Application Number 83339. We thank Dr. Qian Chen for her kind suggestions on statistical analyses. This work is supported by the National Natural Science Foundation of China Regional Innovation and Development Joint Foundation (U23A20408), the National Natural Science Foundation of China (82171698, 82170561, 81300279, 81741067, 82100238), the Program for High-level Foreign Expert Introduction of China (G2022030047L), the Natural Science Foundation for Distinguished Young Scholars of Guangdong Province (2021B1515020003), Guangzhou Basic and Applied Basic Research Scheme-Project for Pilot Voyage (2024A04J6573), the Guangdong Basic and Applied Basic Research Foundation (2022A1515012081), the Foreign Distinguished Teacher Program of Guangdong Science and Technology Department (KD0120220129), the Climbing Program of Introduced Talents and High-level Hospital Construction Project of Guangdong Provincial People's Hospital (DFJH201923, DFJH201803, KJ012019099, KJ012021143, KY012021183), and in part by VA Clinical Merit and ASGE clinical research funds (FWL).

## Additional information

### Funding

| Funder | Grant reference number | Author |
|---|---|---|
| National Natural Science Foundation of China Regional Innovation and Development Joint Foundation | U23A20408 | Hao Chen |
| National Natural Science Foundation of China | 82171698 | Hao Chen |

| Funder | Grant reference number | Author |
|---|---|---|
| National Natural Science Foundation of China | 82170561 | Hao Chen |
| National Natural Science Foundation of China | 81300279 | Hao Chen |
| National Natural Science Foundation of China | 81741067 | Hao Chen |
| National Natural Science Foundation of China | 82100238 | Hao Chen |
| Program for High-level Foreign Expert Introduction of China | G2022030047L | Hao Chen |
| Natural Science Foundation for Distinguished Young Scholar of Guangdong Province | 2021B1515020003 | Hao Chen |
| Guangzhou Basic and Applied Basic Research Scheme | Project for Pilot Voyage 2024A04J6573 | Hao Chen |
| Guangdong Basic and Applied Basic Research Foundation | 2022A1515012081 | Hao Chen |
| Foreign Distinguished Teacher Program of Guangdong Science and Technology Department | KD0120220129 | Hao Chen |
| Climbing Program of Introduced Talents and High-level Hospital Construction Project of Guangdong Provincial People's Hospital | DFJH201923 | Hao Chen |
| Climbing Program of Introduced Talents and High-level Hospital Construction Project of Guangdong Provincial People's Hospital | DFJH201803 | Hao Chen |
| Climbing Program of Introduced Talents and High-level Hospital Construction Project of Guangdong Provincial People's Hospital | KJ012019099 | Hao Chen |
| Climbing Program of Introduced Talents and High-level Hospital Construction Project of Guangdong Provincial People's Hospital | KJ012021143 | Hao Chen |
| Climbing Program of Introduced Talents and High-level Hospital Construction Project of Guangdong Provincial People's Hospital | KY012021183 | Hao Chen |
| American Society for Gastrointestinal Endoscopy | VA Clinical Merit and ASGE clinical research funds | Felix W Leung |

| Funder | Grant reference number | Author |
|---|---|---|

The funders had no role in study design, data collection and interpretation, or the decision to submit the work for publication.

## Author contributions
Ruijie Zeng, Data curation, Formal analysis, Writing - original draft; Yuying Ma, Lijun Zhang, Data curation, Formal analysis; Dongling Luo, Rui Jiang, Huihuan Wu, Zewei Zhuo, Qi Yang, Jingwei Li, Investigation; Felix W Leung, Chongyang Duan, Weihong Sha, Hao Chen, Supervision, Writing – review and editing

## Author ORCIDs
Hao Chen http://orcid.org/0000-0003-4339-3441

Reviewer #1 (Public Review): https://doi.org/10.7554/eLife.94973.3.sa1
Author response https://doi.org/10.7554/eLife.94973.3.sa2

## Additional files

### Supplementary files
• Supplementary file 1. Supplementary information for the analyses. (**a**) Generic name and examples of trade name of proton pump inhibitors. (**b**) Definitions of outcomes in the UK Biobank cohort. (**c**) The proportional hazards assumption tested by Schoenfeld residuals tests. (**d**) Associations of PPI use with the risk of influenza, pneumonia, and other respiratory infections (with inclusion of self-reported cases). (e). Associations of PPI use with COVID-19 severity and mortality. (**f**) Associations of PPI use with the risk of influenza, pneumonia, COVID-19, and other respiratory infections by different types of PPIs. (**g**) Associations of PPI use with the risk of influenza, pneumonia, COVID-19, and other respiratory infections by *CYP2C19* phenotypes (**h**) Associations of PPI use with COVID-19 severity and mortality by *CYP2C19* phenotypes. (**i**) Associations of PPI use with the risk of influenza, pneumonia, COVID-19, and other respiratory infections with multiple imputation. (**j**) Analysis of associations of PPI use with COVID-19 severity and mortality with multiple imputation. (**k**) Clinical characteristics of included participants after propensity score-matching. (**l**) Propensity score-matched analysis of associations of PPI use with the risk of influenza, pneumonia, COVID-19, and other respiratory infections. (**m**) Propensity score-matched analysis of associations of PPI use with COVID-19 severity and mortality. (**n**) Comparisons between proton pump inhibitor (PPI) and histamine-2 receptor antagonist (H2RA) users for COVID-19 severity and mortality.

• MDAR checklist

### Data availability
The current manuscript is a computational study, and therefore no data have been generated for this manuscript. The UK Biobank data are available on application to the UK Biobank (Approval Number 83339). The codes used in this study can be found at: https://epirhandbook.com/en/ and https://cran.r-project.org/doc/contrib/Epicalc_Book.pdf.

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
