## [Editor Report · eLife assessment]

This **useful** study aimed to quantify associations between regular use of proton-pump inhibitors (PPI) with the occurrence of respiratory infections, such as influenza, pneumonia, COVID-19, and others over a period of several years. PPI use was associated with increased risks of influenza, pneumonia, but not of COVID-19, although severity and mortality of COVID-19 infections were higher in PPI users. There are inevitable weaknesses of the study design used, such as the fact that PPI use was only measured at one time-point whereas infections were assessed over a long time period, but these are appropriately highlighted in the discussion. Weaknesses are highlighted in the discussion and the study presents **convincing** evidence for the conclusions overall.

---

## [Referee Report · Reviewer #1 (Public Review)]

Summary:

The current study aims to quantify associations between regular use of proton-pump inhibitors (PPI) - defined as using PPI most days of the week during the last 4 weeks at one cross-section in time - with several respiratory outcomes (6 in total: risk of influenza, pneumonia, COVID-19, other respiratory tract infections, as well as COVID-19 severity and mortality) up to several years later in time.

Strengths:

Several sensitivity analyses were performed, including (i) estimation of the e-value to assess how strong unmeasured confounders should be to explain observed effects, (ii) comparison with another drug with a similar indication to potentially reduce (but not eliminate) confounding by indication, (iii)

Weaknesses:

While the original submission had several weaknesses, the authors have appropriately addressed all issues raised. There are inevitable weaknesses remaining, but these are appropriately highlighted in the discussion. Remaining weaknesses that remain - but are highlighted in the discussion - include the fact that the main exposure of interest is only measured at one time-point whereas outcomes are assessed over a long time period, the inclusion of prevalent users leading to potential bias (e.g. those experiencing bad outcomes already stopping because of side-effects before inclusion in the study), and the possibility of unmeasured confounding explaining observations (e.g. severity of underlying comorbidities leading to PPI prescriptions combined with the absence of information about comorbidity severity), and potential selection bias.

---

## [Author Response]

The following is the authors’ response to the original reviews.

**Reviewer #1 (Public Review):**
Summary:The current study aims to quantify associations between the regular use of proton-pump inhibitors (PPI) - defined as using PPI most days of the week during the last 4 weeks at one cross-section in time - with several respiratory outcomes up to several years later in time. There are 6 respiratory outcomes included: risk of influenza, pneumonia, COVID-19, other respiratory tract infections, as well as COVID-19 severity and mortality.Strengths:Several sensitivity analyses were performed, including (i) estimation of the e-value to assess how strong unmeasured confounders should be to explain observed effects, (ii) comparison with another drug with a similar indication to potentially reduce (but not eliminate) confounding by indication.

We are grateful for your pointing out the strengths in our article, particularly the assessment of e-values and the comparison with another medication to mitigate confounding by indication. We extend our sincere gratitude to the reviewer for identifying multiple concerns and offering constructive feedback to help improve our manuscript. We will incorporate these suggestions into our revisions.

Weaknesses:(1) The main exposure of interest seems to be only measured at one time-point in time (at study enrollment) while patients are considered many years at risk afterwards without knowing their exposure status at the time of experiencing the outcome. As indicated by the authors, PPI are sometimes used for only short amounts of time. It seems biologically implausible that an infection was caused by using PPI for a few weeks many years ago.

We agree with the reviewer that PPIs are sometimes used for only short amounts of time, as indicated in our manuscript. We acknowledge that it is a limitation of the UK Biobank cohort, and we have discussed this in the discussion section as follows:

“Given that the PPI exposure was mainly assessed at the baseline recruitment, it was possible that a small proportion of PPI users was misclassified during the follow-up due to the medication discontinuation, which may result in an underestimation of potential risk.” (Page 14, Line 8-10)

In addition, to alleviate these concerns, we have conducted effect medication for the subgroup of potential long-term users, which were defined by participants with indications of PPI use. This information has been included in the discussion section:

“In addition, no effect moderation was observed in subgroup analyses for the main outcome among PPI users with indications (more likely to regularly use PPIs for a long period) compared to those without indications, indicating the risks remained increased among long-term PPI users.” (Page 14, Line 12-15)

We hope that in the future, the concerns highlighted by the reviewer can be resolved by utilizing datasets with close follow-up, especially regarding medication use:

“Since the follow-up prescription data was lacking in our study to precisely identifying the long-term users, further evaluation using cohorts with close follow-up is needed.” (Page 14, Line 15-17)

(2) Previous studies have shown that by focusing on prevalent users of drugs, one often induces several biases such as collider stratification bias, selection bias through depletion of susceptible, etc.

Because of the limitations of data from the UK Biobank, such as the absence of details on initiation of medications and regular monitoring, we were restricted to using a prevalent user design to assess the associations between PPI use and respiratory outcomes. We have discussed it in the limitation section:

“Given that the PPI exposure was mainly assessed at the baseline recruitment, it was possible that a small proportion of PPI users was misclassified during the follow-up due to the medication discontinuation, which may result in an underestimation of potential risk. However, the prevalent user design could underestimate the actual risks of PPI use for respiratory infections, which indicates the real effect might be stronger [38]……Since the follow-up prescription data was lacking in our study to precisely identifying the long-term users, further evaluation using cohorts with close follow-up is needed.” (Page 14, Line 8-17)

(3) It seems Kaplan Meier curves are not adjusted for confounding through e.g. inverse probability weighting. As such the KM curves are currently not informative (or the authors need to make clearer that curves are actually adjusted for measured confounding).

Your kind suggestions are greatly appreciated. We have plotted Kaplan Meier curves adjusted for confounding by inverse probability weighting with the measured confounders according to the reviewer’s advice. The methods and results are demonstrated as follows:

“The event-free probabilities were compared by Kaplan-Meier survival curves with inverse probability weights adjusting for the measured covariates.” (Page 8, Line 13-15)

“Regular PPI users had lower event-free probabilities for influenza and pneumonia compared to those of non-users (Supplementary Figure 2 A-B).” (Page 9, Line 21-23)

“PPI users had lower event-free probabilities for COVID-19 severity and mortality, but not COVID-19 positivity compared to those of non-users (Supplementary Figure 2 C-E).” (Page 10, Line 9-10)

(4) Throughout the manuscript the authors seem to misuse the term multivariate (using one model with e.g. correlated error terms to assess multiple outcomes at once) when they seem to mean multivariable.

We apologize for misusing the term “multivariate” and “multivariable” in our previous manuscript. We have corrected the misused terms throughout the manuscript:

“Univariate and multivariable Cox proportional hazards regression models were utilized to assess the association between regular use of PPIs and the selected outcomes.” (Page 7, Line 19-20)

“The remaining imbalanced covariates (standardized mean difference ≥ 0.1) after propensity score matching were further adjusted by multivariate multivariable Cox regression models to calculate HRs and 95% CIs.” (Page 8, Line 23-25)

(5) Given multiple outcomes are assessed there is a clear argument for accounting for multiple testing, which following the logic of the authors used in terms of claiming there is no association when results are not significant may change their conclusions. More high-level, the authors should avoid the pitfall of stating there is evidence of absence if there is only an absence of evidence in a better way (no statistically significant association doesn't mean no relationship exists).

We have revised our interpretation for the results, particularly for those without statically significant association based on the reviewer’s advice, and clearly recognize that the conclusions should be interpreted with cautions:

“In contrast, the risk of COVID-19 infection was not significant with regular PPI use…” (Page 2, Line 11-12)

“PPI users were associated with a higher risk of influenza (HR 1.74, 95%CI 1.19-2.54), but the risks with pneumonia or COVID-19-related outcomes were not evident.” (Page 2, Line 14-16)

“…while the effects on pneumonia or COVID-19-related outcomes under PPI use were attenuated when compared to the use of H2RAs.” (Page 2, Line 18-19, in the Abstract)

“…while their association with pneumonia and COVID-19-related outcomes is diminished after comparison with H2RA use and remains to be further explored.” (Page 15, Line 21-22, in the Conclusion)

(6) While the authors claim that the quantitative bias analysis does show results are robust to unmeasured confounding, I would disagree with this. The e-values are around 2 and it is clearly not implausible that there are one or more unmeasured risk factors that together or alone would have such an effect size. Furthermore, if one would use the same (significance) criteria as used by the authors for determining whether an association exists, the required effect size for an unmeasured confounder to render effects 'statistically non-significant' would be even smaller.

We agree with the reviewer that there might still exist one or more unmeasured risk factors that have effect sizes larger than 2. Hence, we cannot affirm that the findings are robust to unmeasured confounding in the current analysis, which is a limitation of our study. We have deleted the previous statement, and added more discussion in the limitation section:

“Moreover, patients with exacerbations of respiratory disorders (e.g., asthma, COPD) might suffer from a wide range of gastrointestinal symptoms that lead to the use of PPIs [38]. Due to the lack of data for respiratory severity and close follow-up for medication use, residual confounding might still exist due to the observational nature.” (Page 14, Line 23-27)

(7) Some patients are excluded due to the absence of follow-up, but it is unclear how that is determined. Is there potentially some selection bias underlying this where those who are less healthy stop participating in the UK biobank?

Thank you for your question. The reasons for the absence of follow-up are mainly classified into five categories, including: (1) Death reported to UK Biobank by a relative; (2) NHS records indicate they are lost to follow-up; (3) NHS records indicate they have left the UK; (4) UK Biobank sources report they have left the UK; (5) Participant has withdrawn consent for future linkage. According to the data from UK Biobank (https://biobank.ndph.ox.ac.uk/showcase/field.cgi?id=190), the major reason for the loss of follow-up among participants is their departure from the UK (84.7% of participants who were lost to follow-up). In addition, not including those who were less healthy in the study might also underestimate the risk, leading to lower estimated effects of PPIs for respiratory infections. We have supplemented this in our revised manuscript:

“Among them, 1,297 participants without follow-up, which were mainly determined by reported death, departure from the UK, or withdrawn consent, had been removed after initial exclusion.” (Page 4, Line 25-27)

(8) Given that the exposure is based on self-report how certain can we be that patients e.g. do know that their branded over-the-counter drugs are PPI (e.g. guardium tablets)? Some discussion around this potential issue is lacking.

Thank you for your concerns. In the data collection by the UK Biobank, the participants can enter the generic or trade name of the treatment on the touchscreen to match the medications they used. We have added this important information to the method section:

“The exposure of interest was regular use of PPIs. The participants could enter the generic or trade name of the treatment on the touchscreen to match the medications they used (Supplementary Table S1).” (Page 5, Line 6-8)

We acknowledge that specific information on prescribed or over-the-counter use of medications is lacking in the UK Biobank. We have discussed it in the limitation section:

“Limitations exist in our study. Information on dose and duration of PPI use, discrimination between prescription and over-the-counter use of PPIs, health-seeking behavior, different types of pneumonia, and pneumococcus vaccination is currently not available from the UK Biobank.” (Page 14, Line 5-8)

(9) Details about the deprivation index are needed in the main text as this is a UK-specific variable that will be unfamiliar to most readers.

Thank you for your question on the definition of deprivation index. We have proved the details about the deprivation index in the manuscript:

“…socioeconomic status (deprivation index, which was defined using national census information on car ownership, household overcrowding, owner occupation, and unemployment combined for postcode areas of residence)…” (Page 6, Line 14-17)

(10) It is unclear how variables were coded/incorporated from the main text. More details are required, e.g. was age included as a continuous variable and if so was non-linearity considered and how?

We apologize for not elucidating how variables were incorporated into the main text. Previously, the linearity between continuous variables and outcomes was assessed by Martingale residuals plots, while the variables detected with non-linearity were regarded as categorical variables for further analyses. For example, after evaluation with the Martingale residuals plot, age demonstrated non-linearity, and we incorporated it as a categorical variable for the analysis of COVID-related mortality.

We have supplemented the information in the method section:

“The linearity between continuous variables and outcomes was assessed by Martingale residuals plots, while the variables detected with non-linearity were regarded as categorical variables for further analyses.” (Page 6, Line 28 to Page 7, Line 1)

(11) The authors state that Schoenfeld residuals were tested, but don't report the test statistics. Could they please provide these, e.g. it would already be informative if they report that all p-values are above a certain value.

We are sorry for not providing the statistics about the Schoenfeld residual in our previous manuscript. We have supplemented the information in our revisions:

“Schoenfeld residuals tests were used to evaluate the proportional hazards assumptions, while no violation of the assumption was detected (Supplementary Table S3).” (Page 7, Line 27 to Page 8, Line 1)

(12) The authors would ideally extend their discussion around unmeasured confounding, e.g. using the DAGs provided in https://www.ncbi.nlm.nih.gov/pmc/articles/PMC7832226/, in particular (but not limited to) around severity and not just presence/absence of comorbidities.

Thank you for your insightful suggestions that the discussion about unmeasured confounding should be extended. We agree with the reviewer that, in addition to the comorbidities themselves, their severity could also have an important impact on the use of PPIs. We have added the discussion in the limitation section with citing the article (PMC7832226):

“Moreover, patients with exacerbations of comorbid disorders (e.g., diabetes, asthma, COPD) might suffer from a wide range of gastrointestinal symptoms that lead to the use of PPIs [38] (Supplementary Figure S4). Due to the lack of data for respiratory severity and close follow-up for medication use, residual confounding might still exist due to the observational nature.” (Page 14, Line 23-27)

(13) The UK biobank is known to be highly selected for a range of genetic, behavioural, cardiovascular, demographic, and anthropometric traits. The potential problems this might create in terms of collider stratification bias - as highlighted here for example: https://www.nature.com/articles/s41467-020-19478-2 - should be discussed in greater detail and also appreciated more when providing conclusions.

We acknowledge the reviewer's point about the UK Biobank's highly selective nature potentially leading to collider stratification bias in the evaluation of COVID-19-related outcomes. We have discussed this in detail and are cautious when generating conclusions.

“Furthermore, the highly selective nature of the UK Biobank might create collider stratification bias for the evaluation of COVID-19-related outcomes, and thus the conclusions should be interpreted with cautions [39].” (Page 15, Line 2-4)

**Reviewer #2 (Public Review):**
Summary:Zeng et al investigate in an observational population-based cohort study whether the use of proton pump inhibitors (PPIs) is associated with an increased risk of several respiratory infections among which are influenza, pneumonia, and COVID-19. They conclude that compared to non-users, people regularly taking PPIs have increased susceptibility to influenza, pneumonia, as well as COVID-19 severity and mortality. By performing several different statistical analyses, they try to reduce bias as much as possible, to end up with robust estimates of the association.Strengths:The study comprehensively adjusts for a variety of critical covariates and by using different statistical analyses, including propensity-score-matched analyses and quantitative bias analysis, the estimates of the associations can be considered robust.

We are grateful to the reviewer for pointing out the merits of our articles, which include adjusting for a wide range of covariates, employing diverse statistical analyses, and using robust data. We will revise our manuscript further based on the reviewer's suggestions.

Weaknesses:As it is an observational cohort study there still might be bias. Information on the dose or duration of acid suppressant use was not available, but might be of influence on the results. The outcome of interest was obtained from primary care data, suggesting that only infections as diagnosed by a physician are taken into account. Due to the self-limiting nature of the outcome, differences in health-seeking behavior might affect the results.

Thank you for your questions for information on the dose/duration of acid suppressants, the source of diagnosis, and the health-seeking behavior of participants. For the data from the UK Biobank, the dose or duration of acid suppressant use was not available since the information was not collected as baseline or follow-up. In addition, the outcome of interest was also retrieved from the hospital ICD diagnosis. We apologize for not clarifying it in our previous manuscript. Moreover, we agree with the reviewer that the health-seeking behavior could have an impact on the analyses, whereas the correlated data are still not available from the UK Biobank. We have discussed them in the method and limitation section:

“Briefly, the first reported occurrences of respiratory system-related conditions within primary care data, and hospital inpatient data defined by the International Classification of Diseases (ICD)- 10 codes were categorized by the UK Biobank.” (Page 5, Line 21-25)

“Limitations exist in our study. Information on dose and duration of PPI use, discrimination between prescription and over-the-counter use of PPIs, health-seeking behavior, different types of pneumonia, and pneumococcus vaccination is currently not available from the UK Biobank.” (Page 14, Line 5-8)

**Reviewer #1 (Recommendations For The Authors):**
Analysis code should be made available.

Thank you for your question. We have provide the sources of the analysis code we used for this study in our revised manuscript:

“The codes used in this study can be found at: https://epirhandbook.com/en/ and https://cran.r-project.org/doc/contrib/Epicalc_Book.pdf.” (Page 16, Line 21-22)

**Reviewer #2 (Recommendations For The Authors):**

It might be interesting to study whether including self-reported infections changes the results, as people using PPI may more easily consult their GP even for a self-limiting disease such as influenza and therefore are more likely diagnosed/confirmed with such a respiratory infection.

Thank you for your insightful suggestions on conducting analyses including self-reported infections. Therefore, we have included the self-reported cases as sensitivity analyses, and the results were not significantly altered, which confirms the robustness of our results:

“Self-reported infections, except for COVID-19-related outcomes due to the lack of data, were also included for the outcomes as sensitivity analyses. The self-reported cases were reported at the baseline or subsequent UK Biobank assessment center visit.” (Page 8, Line 17-19)

“Inclusion of the self-reported cases did not significantly alter the results (Supplementary Table S4).” (Page 9, Line 17-18)

Moreover, to address the above-mentioned, sub-analyses differentiating between over-the-counter and prescribed medication might be interesting.

Thank you for your questions on differentiating between over-the-counter and prescribed medication. We have thoroughly looked up the data provided by the UK Biobank, but it is a pity that they are not provided. We have discussed this in the limitation section:

“Information on dose and duration of PPI use, discrimination between prescription and over-the-counter use of PPIs, health-seeking behavior, different types of pneumonia, and pneumococcus vaccination is currently not available from the UK Biobank.” (Page 14, Line 5-8)